# Glucuronidated Metabolites of Bisphenols A and S Alter the Properties of Normal Urothelial and Bladder Cancer Cells

**DOI:** 10.3390/ijms232112859

**Published:** 2022-10-25

**Authors:** Ève Pellerin, Félix-Antoine Pellerin, Stéphane Chabaud, Frédéric Pouliot, Martin Pelletier, Stéphane Bolduc

**Affiliations:** 1Centre de Recherche en Organogénèse Expérimentale/LOEX, Regenerative Medicine Division, CHU de Québec-Université Laval Research Center, Québec, QC G1J 1Z4, Canada; 2Oncology Division, CHU de Québec-Université Laval Research Center, Québec, QC G1R 2J6, Canada; 3Department of Surgery, Faculty of Medicine, Laval University, Québec, QC G1V 0A6, Canada; 4Infectious and Immune Disease Division, CHU de Québec-Université Laval Research Center, Québec, QC G1V 4G2, Canada; 5Department of Microbiology-Infectious Diseases and Immunology, Faculty of Medicine, Laval University, Québec, QC G1V 0A6, Canada

**Keywords:** bladder cancer, metabolites, bisphenol A glucuronide, bisphenol S glucuronide, energy metabolism, migration, proliferation, urothelium

## Abstract

Bisphenol A (BPA) and bisphenol S (BPS) are synthetic chemicals used to produce plastics which can be released in food and water. Once ingested, BPA and BPS are metabolized by the liver, mainly as glucuronidated metabolites, and are excreted through urine. Since urine can be stored for many hours, the bladder is chronically exposed to BP metabolites, and studies have shown that these metabolites can remain active in the organism. Therefore, the effect of physiological concentrations of glucuronidated BPs was evaluated on the bioenergetics (glycolysis and mitochondrial respiration), migration and proliferation of normal urothelial cells, and non-invasive and invasive bladder cancer cells. The results demonstrated that an exposure of 72 h to glucuronidated BPA or BPS decreased the bioenergetics and activity of normal urothelial cells, while increasing these parameters for bladder cancer cells. These findings suggest that BP metabolites are not as inactive as initially believed, and their ubiquitous presence in the urine could promote bladder cancer progression.

## 1. Introduction

Bisphenol A (BPA) and bisphenol S (BPS) are synthetic chemicals used to produce plastics [1]. These compounds can be released in food and water, thus resulting in the continuous exposure of humans to BPs [2]. BPA and BPS are characterized as endocrine disruptors since they can bind to multiple cell receptors, such as estrogen receptors (ERs), the androgen receptor (AR), and the G protein-coupled estrogen receptor (GPER), resulting in the modulation of signalling pathways associated with migration, proliferation, and invasion [2,3]. In addition, chronic exposure to these compounds has been associated with cancer development, especially for hormone-dependent breast [4,5] and prostate cancers [6,7].

Once ingested and absorbed by the organism, BPA and BPS are usually rapidly metabolized by the liver. BPA and BPS are then conjugated to produce metabolites. In fact, 99% of absorbed BPA is metabolized by the liver. In comparison, only 41% of ingested BPS is converted to metabolites due to its reduced excretion compared to BPA, thus resulting in an increased bioavailability [8]. The primary metabolites produced are obtained following the conjugation of BPA or BPS with glucuronic acid, which respectively makes BPA-glucuronide (BPA-gluc) (69.5%) or BPS-glucuronide (BPS-gluc) (85.8%) [1]. The second most common metabolite is obtained following the conjugation of BPs with sulphate (21.0% of BPA and 10.5% of BPS) [1]. Both glucuronidated and sulphated metabolites are mainly excreted through urine [9]. Urine can be stored for many hours before evacuation outside the bladder, leading to the bladder’s chronic exposure to BPs and their metabolites [10]. Studies by Tang et al. have shown that certain aromatic amines found in tobacco smoke and industrial chemicals can be conjugated with a UDP-glucuronic acid by UDP-glucuronosyltransferases (UGTs) [11]. This conjugation allows the excretion of these compounds through urine. However, the acidity of the urine (pH < 6.0) can cause instability in glucuronide metabolites, thus resulting in the dissociation and release of the original aromatic amines [11]. Although this phenomenon has not been shown with glucuronidated metabolites, it can be hypothesized that, in certain circumstances, glucuronidated BPA and BPS could be dissociated and released back to their original form in the urine. BPA and BPS were found at similar concentrations in the urine of adults in the United States, ranging from 1 to 100 nM [3], and studies have shown that glucuronidated metabolites are present in the urine at comparable concentrations to BPA. Harthé et al. measured BPA-gluc concentration in the urine of 163 hospital patients in France and established its concentration at an average of 10 nM [12]. Similarly, Trabert et al. measured the presence of BPA-gluc in urine samples of 575 postmenopausal women in Poland and also established its concentration at an average of 10 nM [13]. Although urinary concentrations of BPS-gluc have not been established, the presence of BPS-gluc in urine at a comparable concentration to BPA-gluc can be expected.

It was believed that BPA-gluc was inactive for a long time because of its poor binding capacity to ERs [14]. Matthews et al. showed that BPA-gluc did not have a competitive binding site to ERs and did not activate ERα and ERβ compared to BPA [15]. These results suggest that BPA-gluc has no estrogenic activity [15]. Nonetheless, Boucher et al. have shown that ER antagonists inhibited the induction of adipocyte differentiation by BPA-gluc. However, BPA-gluc did not show any estrogenic activity, which could suggest the use of an alternative pathway [14]. Studies have shown that BPS-gluc was inactive for estrogenic and androgenic activity in yeast cells and had no inhibiting activity on the ERs [16]. Moreover, studies by Peillex et al. have shown that the exposure of human neutrophils to BPS-gluc or BPA-gluc altered their metabolism and anti-microbial functions [17]. These studies demonstrate that BP metabolites can remain active in the organism and impact various cellular processes.

It was previously demonstrated that chronic exposure to physiological concentrations of BPA and BPS decreased the bioenergetics and several biological properties of normal urothelial cells, while increasing these parameters for non-invasive and invasive bladder cancer cells [18]. Since BPs metabolites are also found in urine, we hypothesized that chronic exposure to physiological concentrations of BPA-gluc or BPS-gluc would have similar effects on the energy metabolism and properties of urothelial and bladder cancer cells than their original compound. Therefore, it could potentially impact bladder cancer progression. Consequently, we evaluated the impact of chronic exposure to physiological concentrations of glucuronidated BPs on the bioenergetics, migration and proliferation of normal urothelial cells and non-invasive and invasive bladder cancer cells.

## 2. Results

### 2.1. Chronic Exposure to Physiological Concentrations of Glucuronidated BPS Decreases the Basal Glycolytic Capacity of Normal Urothelial Cells

Since BP metabolites are mainly excreted through urine [9], bladder urothelial cells will likely be chronically exposed to these compounds. Two populations of normal urothelial cells (UCs) were exposed to physiological concentrations of BPA-gluc or BPS-gluc to evaluate their impact on energy metabolism (Appendix A). A 72 h exposure to 10^−8^ M BPA-gluc tended to decrease the basal glycolysis of UCs (*p* = 0.075), while 10^−8^ M BPS-gluc significantly reduced the basal glycolysis of UCs (Figure 1A). BPA-gluc and BPS-gluc also tended to decrease the maximal glycolytic capacity of UCs (Figure 1B). However, chronic exposure to glucuronidated BPs had no impact on the basal (Figure 1C) and maximal (Figure 1D) mitochondrial respiration of UCs. Therefore, UCs chronically exposed to physiological concentrations of BPA-gluc and BPS-gluc generally exhibited a decreased glycolytic metabolism.

### 2.2. RT4 Non-Invasive Bladder Cancer Cells Chronically Exposed to Physiological Concentrations of BPA and BPS Glucuronidated Metabolites Exhibit Increased Bioenergetics

Like UCs, bladder cancer cells can be exposed to glucuronidated metabolites through urine stored in the bladder. RT4 non-invasive bladder cancer cells were chronically exposed to physiological concentrations of BPA-gluc or BPS-gluc to evaluate the impact on their bioenergetics (Appendix A). Chronic exposure to 10^−8^ M BPA-gluc or BPS-gluc significantly increased the basal (Figure 2A) and maximal (Figure 2B) glycolytic capacity of RT4 cells. Furthermore, RT4 cells chronically exposed to 10^−8^ M BPA-gluc exhibited increased basal (Figure 2C) and maximal (Figure 2D) mitochondrial respiration. Exposure to 10^−8^ M BPS-gluc did not significantly impact the mitochondrial metabolism of RT4 cells, although we observed an increase in basal and maximal mitochondrial respiration (Figure 2C,D). Thus, chronic exposure to physiological concentrations of BPA-gluc or BPS-gluc increased the energy metabolism of RT4 non-invasive cancer cells.

### 2.3. T24 Invasive Bladder Cancer Cells Chronically Exposed to Physiological Concentrations of Glucuronidated BPS Exhibit an Increased Maximal Glycolytic Capacity

T24 invasive bladder cancer cells were also chronically exposed to physiological concentrations of BPA-gluc or BPS-gluc to establish their impact on cellular energy metabolism (Appendix A). Chronic exposure to 10^−8^ M BPA-gluc did not impact the basal glycolysis (Figure 3A), while 10^−8^ M BPS-gluc tended to increase the basal glycolysis of T24 cells (*p* = 0.096). In addition, T24 cells exposed to 10^−8^ M BPA-gluc tended to exhibit an increased maximal glycolytic capacity (*p* = 0.054) (Figure 3B), while chronic exposure to 10^−8^ M BPS-gluc significantly increased the maximal glycolytic capacity of T24 cells (Figure 3B). BPA-gluc and BPS-gluc exposure did not impact the basal (Figure 3C) and maximal (Figure 3D) mitochondrial respiration of T24 cells. Therefore, chronic exposure to physiological concentrations of BPS-gluc increased the maximal glycolytic capacity of T24 invasive bladder cancer cells.

### 2.4. Chronic Exposure to Physiological Concentrations of Glucuronidated BPA and BPS Decreases the Migration Speed of Normal Urothelial Cells, While Glucuronidated BPA Tends to Increase the Migration of T24 Invasive Bladder Cancer Cells

The migration of UCs, RT4 and T24 cells was evaluated following chronic exposure to physiological concentrations of BPA-gluc and BPS-gluc. UCs chronically exposed to 10^−8^ M BPA-gluc or BPS-gluc exhibited a significantly decreased migration compared to the control (Figure 4A). However, exposure to glucuronidated BPs did not impact the migration of non-invasive RT4 cells (Figure 4B). Finally, chronic exposure to 10^−8^ M BPA-gluc tended to increase the migration of T24 cells (*p* = 0.065), but exposure to BPS-gluc did not affect the migration of these cells (Figure 4C). Overall, chronic exposure to physiological concentrations of BPA-gluc or BPS-gluc decreased the migration of UCs, while BPA-gluc tended to increase the migration of T24 invasive bladder cancer cells.

### 2.5. UCs and RT4 Non-Invasive Bladder Cancer Cells Chronically Exposed to Physiological Concentrations of Glucuronidated BPA and BPS Exhibit an Increased Proliferation

The proliferation rate of UCs, RT4 and T24 cells was evaluated for three days under chronic exposure to physiological concentrations of BPA-gluc or BPS-gluc. First, chronic exposure to 10^−8^ M BPA-gluc or BPS-gluc significantly increased the proliferation of UCs on days 1 to 3 (Figure 5A). Furthermore, UCs exposed to 10^−8^ M BPA-gluc exhibited a significantly increased proliferation rate compared to UCs exposed to BPS-gluc on day 1. However, this difference was not maintained over time. Secondly, chronic exposure to 10^−8^ M BPA-gluc or BPS-gluc significantly increased the proliferation rate of RT4 cells on day 3 (Figure 5B). Additionally, RT4 cells exposed to BPA-gluc tended to have an increased proliferation rate on day 3 compared to RT4 cells exposed to BPS-gluc (*p* = 0.094). Thirdly, exposure to glucuronidated BPs did not impact the proliferation rate of T24 invasive bladder cancer cells (Figure 5C). Overall, chronic exposure of UCs and RT4 non-invasive bladder cancer cells to physiological concentrations of BPA-gluc or BPS-gluc increased their proliferation rate.

## 3. Discussion

In vivo, UCs can be exposed to BP metabolites through urine. However, the basal layer of UCs is usually protected from the contaminants found in urine, such as urea and BPs. Still, specific pathologies such as bladder cancer and urinary infection can alter the urothelium’s impermeability [19,20]. For example, in the case of a urinary infection, the bladder can induce the shedding of its superficial urothelial layer to reduce the bacterial load, exposing the underlying UCs to urine and its contaminants [20]. Furthermore, in the case of bladder cancer, cancer cell growth can disrupt cell-cell adhesion [19], consequently exposing the underlying urothelial layers and UCs surrounding the tumour to urine and potentially BPs and their glucuronidated metabolites. Furthermore, UCs can be exposed to BP metabolites through the vascular system perfusing the bladder [21]. Therefore, the impact of chronic exposure to physiological concentrations of BPA-gluc or BPS-gluc on normal UCs and RT4 non-invasive and T24 invasive bladder cancer cells was studied. In addition, the consequences of glucuronidated BP exposure on energy metabolism, migration, and proliferation were examined.

First, UCs chronically exposed to BPA-gluc or BPS-gluc exhibited a decreased glycolytic metabolism, while the mitochondrial respiration remained unchanged. These results could be explained by the inhibition of metabolizing enzymes by BPs. Although no studies have directly evaluated the impact of glucuronidated BPs on enzyme activity, it is known that BPA can inhibit certain enzymes associated with energy metabolism, like enzymes from the electron transport chain and glucose transporters [22]. This BPA-induced enzymatic activity alteration can affect major metabolic pathways, such as mitochondrial respiration [22,23]. Therefore, it is possible to hypothesize that glucuronidated BPs could inhibit key enzymes in glycolysis, which could explain the decreased glycolytic capacity observed in UCs after exposure to BP metabolites. Furthermore, chronic exposure to BPA-gluc or BPS-gluc reduced the migration of UCs, which could affect the ability of the urothelium to repair itself in case of injury. Since cell migration is crucial for wound closure, a decreased migration of UCs could lead to slower healing of the bladder wall, thus increasing the exposure of underlying cell layers to urinary BP metabolites [12,13,24]. The decreased migration could also result from reduced bioenergetics. However, UCs chronically exposed to BPA-gluc or BPS-gluc exhibited an increased proliferation rate at 24, 48, and 72 h. BPA-gluc also had a greater effect on UCs’ proliferation after 24 h, but this difference was not maintained over time. Studies have shown that BPA-gluc could activate the extracellular signal-regulated kinase (ERK) pathway, which is associated with proliferation [25]. A similar mechanism could likely occur with BPS-gluc but remains to be determined.

Secondly, RT4 non-invasive cancer cells chronically exposed to BPA-gluc or BPS-gluc exhibited increased glycolytic and mitochondrial capacities. A consequence of an enhanced glycolytic metabolism is the excessive production of lactate by the cell [26]. Therefore, in vivo exposure of non-invasive bladder tumours to BPA-gluc or BPS-gluc could result in the acidification of the microenvironment, thus inhibiting local immune cells [26,27]. Furthermore, the increased bioenergetics following chronic exposure to glucuronidated BPs could be related to the enhanced physiological activity of RT4 cells. Although chronic exposure to BPA-gluc or BPS-gluc did not affect the migration of non-invasive RT4 cells, it impacted cell proliferation. In fact, RT4 cells tend to stay in clusters and exhibit low migration capacities [18]. Since RT4 cells do not tend to migrate, the effects of glucuronidated BPs could remain unnoticed in vitro. However, non-invasive RT4 cells exposed to physiological concentrations of BPA-gluc or BPS-gluc exhibited an increased proliferation rate after 72 h. This enhancement could be associated with the activation of the ERK pathway [25]. The increased bioenergetics and proliferation of non-invasive RT4 cells exposed to BP metabolites could suggest a more aggressive phenotype, thus promoting the transition from non-invasive to invasive bladder cancer.

Third, T24 invasive cancer cells chronically exposed to glucuronidated BPs exhibited an increased glycolytic metabolism, while the mitochondrial respiration remained unchanged. Similar to RT4 cells, the increased glycolytic capacity of invasive T24 cells leads to higher lactate production, thus resulting in the acidification of the microenvironment, which can inhibit immune cells [26,27]. Chronic exposure to physiological concentrations of BPA-gluc tended to increase the migration of invasive T24 cells, while BPS-gluc did not impact this parameter. Studies have shown that BPA can promote the migration of colon and ovarian cancer cells through the activation of the ERK pathway [28,29]. Therefore, the activation of ERK by BPA-gluc [25] could explain the increased migration of T24 cells. However, exposure to glucuronidated BPs did not affect the proliferation rate of the already highly proliferative invasive T24 cells. These cells’ high proliferation rate could have obscured the impact of BP metabolites on proliferation. Although T24 cells are invasive cancer cells, the increased glycolytic metabolism and migration by glucuronidated BPs’ exposure could potentially speed up metastasis formation.

Of note, using plastic-based laboratory equipment leads to the inability to ensure a BP-free control [30]. Although this limitation must be considered, significant differences were observed between the controls and those treated with glucuronidated BPs. Furthermore, an in vitro 3D model would allow a better representation of the effects of BP metabolites on the tumour microenvironment rather than a 2D cell culture. Finally, the impact of chronic exposure to glucuronidated BPs could eventually be evaluated in a more complex environment using, for example, our 3D bladder cancer model [31].

## 4. Materials and Methods

### 4.1. Cell Lines

Cell extraction and cell culture were conducted as previously described [18]. All procedures involving patients were performed according to Helsinki’s Declaration and were approved by the local Research Ethical Committee. Each sample was obtained with the donor’s consent for tissue harvesting, and all experimental procedures were conducted according to the CHU de Québec-Université Laval guidelines. Normal urothelial cells (UCs) were extracted from healthy human urological tissue biopsies and cultured as previously described [32,33]. In addition, the UCs were isolated from two healthy pediatric volunteers undergoing surgery for a benign condition (UC1 and UC2) and used as non-transformed primary cell lines.

UCs, RT4 non-invasive bladder cancer cells (ATCC HTB-2) and T24 invasive bladder cancer cells (ATCC HTB-4) were maintained in culture media composed of a 3:1 mix of Dulbecco-Vogt modification of Eagle’s medium (DMEM) (Invitrogen, Burlington, ON, Canada) and Ham’s F12 medium (Invitrogen) supplemented with 5% fetal bovine serum clone II (Hyclone, GE Healthcare Life Science, Wauwatosa, WI, USA), 24.3 µg/mL adenine (Corning, Tewksbury, MA, USA), 10 ng/mL epidermal growth factor (Austral Biologicals, San Ramon, CA, USA), 25 mg/mL gentamicin (Schering-Plough Canada Inc./Merck, Scarborough, ON, Canada), 0.4 mg/mL hydrocortisone (Calbiochem, San Diego, CA, USA), 5 µg/mL insulin (Sigma-Aldrich, Oakville, ON, Canada), 0.212 µg/mL isoproterenol (Sandoz, Boucherville, QC, Canada) and 100 U/mL penicillin (Sigma-Aldrich), and incubated at 37 °C with 8% CO_2_. Media were changed three times per week.

### 4.2. Seahorse Energy Metabolism Measurements

Bioenergetics measurements were conducted as previously described [18]. UCs, RT4 and T24 cells were plated in 96-well Seahorse XF cell culture plates (Agilent/Seahorse Bioscience, Chicopee, MA, USA) and exposed or not to 10^−8^ M bisphenol A β-D-glucuronide (BPA-gluc; Toronto Research Chemicals, North York, ON, Canada) or 10^−8^ M 4,4′-bisphenol S O-β-D-glucuronide (BPS-gluc; Toronto Research Chemicals, North York, ON, Canada) 72 h before measurements. Media were changed every day. Seahorse XFe96 sensor cartridge plates (Agilent/Seahorse Bioscience) were hydrated with the XF Calibrant (Agilent/Seahorse Bioscience) the day before the analysis, and incubated overnight at 37 °C without CO_2_. Before the bioenergetics measurements, cells were washed and incubated for 1 h with Glyco Stress media or Mito Stress media. Glyco Stress media contained XF Base Medium (minimal DMEM) (Agilent/Seahorse Bioscience) supplemented with 2 mM L-glutamine (Wisent Bioproducts Inc., Saint-Jean-Baptiste, QC, Canada). Mito Stress media contained XF Base Medium supplemented with 2 mM L-glutamine, 1 mM sodium pyruvate (Wisent Bioproducts Inc.) and 10 mM D-(+)-glucose (Millipore Sigma, Oakville, ON, Canada). The extracellular acidification rate (ECAR), representative of the glycolytic capacity, and the oxygen consumption rate (OCR), representative of the mitochondrial respiration, were determined using the XFe Extracellular Flux Analyzer (Agilent/Seahorse Bioscience) [34,35].

The glycolytic metabolism was determined by the sequential injection of 10 mM D-(+)-glucose (Millipore Sigma), 1.5 µM of the ATP synthase inhibitor oligomycin (Cayman Chemical, Ann Arbor, MI, USA) to inhibit mitochondrial respiration and force the cells to maximize their glycolytic capacity, and 50 mM 2-deoxy-D-glucose (2-DG) (Alfa Aesar, Ward Hill, MA, USA), a competitive inhibitor of the first step of glycolysis.

The mitochondrial respiration was established by the sequential injection of 1.5 µM of the ATP synthase inhibitor oligomycin (Cayman Chemical), 0.5 µM of the mitochondrial uncoupler trifluoromethoxy carbonylcyanide phenylhydrazone (FCCP) (Cayman Chemical), and a combination of 0.5 µM of the mitochondrial complex I inhibitor rotenone (MP Biomedicals, Santa Ana, CA, USA) and 0.5 µM of the mitochondrial complex III inhibitor antimycin A (Millipore Sigma). The concentrations indicated for each injection represent the final concentrations in the wells. At least three measurement cycles (3 min of mixing + 3 min of measuring) were completed before and after each injection.

The OCR and ECAR were calculated using Wave software v2.6 (Agilent/Seahorse Bioscience). Following the manufacturer’s instructions, energy metabolism was normalized according to cell number using a CyQuant Cell proliferation assay kit (Invitrogen). The fluorescence of each well was measured at 485 nm/535 nm for 0.1 s using the Victor2 1420 MultiLabel Counter plate reader (Perkin Elmer Life Sciences, Waltham, MA, USA) and Wallac 1420 software (Perkin Elmer). The normalization values were calculated from the fluorescence measurements with Microsoft Excel software (Microsoft, Redmond, WA, USA) and applied to the metabolic values. Metabolic values were presented as percentages, with 100% established using the first three measurements. Therefore, the baseline was established before the injection of glucose or oligomycin (see Appendix A). Each experiment included at least three replicates per condition (n ≥ 3), and each experiment was repeated at least three times (N ≥ 3).

### 4.3. Migration

Cell migration was established as previously described [18]. UCs, RT4 and T24 cells were seeded in 12-well culture plates at 150,000 cells/well density. Cells were incubated for two hours at 37 °C with 8% CO_2_ to allow cell adhesion. Then, cells were treated or not with 10^−8^ M BPA-gluc or 10^−8^ M BPS-gluc and incubated at 37 °C with 8% CO_2_. After 72 h, a scratch test was performed [36]. Briefly, a vertical scratch was performed in each well using a 200 µL pipet tip. The wells were rinsed twice to remove detached cells with 3:1 DMEM-Ham F12 medium supplemented with 0.5% fetal bovine serum clone II. Two mL of medium supplemented with 0.5% serum with or without 10^−8^ M BPA-gluc or BPS-gluc was added to each well. The low serum concentration avoids cell proliferation and ensures the observed results are due to cell migration. Cell migration was assessed using a Zeiss Axio Imager M2 Time-Lapse microscope equipped with an AxioCam ICc1 camera (Carl Zeiss, Oberkochen, Germany). Images were processed with the AxioVision 40 V4.8.2.0 software (Carl Zeiss). Photographs were taken every hour for a total of 17 h. Migration analyses were measured using ImageJ software (NIH, Bethesda, MD, USA). Migration speed was calculated as the slope of the closure area (Y-axis) as a function of time (X-axis) with the formula “ax + b”, where “a” represents migration speed. The slopes of each cell line were established as percentages of control (i.e., untreated condition). Each condition included two replicates (n = 2) for every cell line, and each experiment was repeated independently three times (N = 3).

### 4.4. Proliferation

The proliferation rate was established as previously described [18]. On day 0, UCs, RT4 and T24 cells were seeded in 24-well culture plates at 60,000 cells/well density. Cells were incubated for two hours at 37 °C with 8% CO_2_ to allow cell adhesion and then treated or not with 10^−8^ M BPA-gluc or 10^−8^ M BPS-gluc. The medium, supplemented or not with BPs, was changed daily for three days. On days 1 to 3, cells from three wells were collected using trypsin, centrifuged at 300 g for 10 min, resuspended in 10 mL ISOTON II diluent (Beckman Coulter, Mississauga, ON, Canada) and counted separately using the Z2 Coulter Particle Count and Size Analyzer (Beckman Coulter) [37]. A graph illustrating the number of cells per well as a function of time was performed to calculate the proliferation rate. Proliferation values for days 1 to 3 were established as percentages of control (i.e., untreated condition) on day 1. Therefore, the proliferation value of the control on day 1 was established as 100%. Each condition included three replicates (n = 3) for every cell line, and each experiment was repeated independently three times (N = 3).

### 4.5. Statistical Analysis

Graphical representation and statistical analyses were performed using Microsoft Excel (Microsoft) and GraphPad Prism Software v.9.3 (San Diego, CA, USA). The results are expressed as mean ± standard error of the mean (SEM). Statistical analyses were performed using the non-parametric Mann–Whitney test. The values did not meet the required premises to assume normality, which justifies using non-parametric statistical tests. Statistical significance was established at *p* < 0.05.

## 5. Conclusions

Despite the presence of glucuronidated BPs in the urine, the impact of chronic exposure to these metabolites on bladder cancer had not yet been evaluated. This study revealed the consequences of BPA-gluc and BPS-gluc exposure on normal urothelial and bladder cancer cells. The effects of chronic exposure to BP metabolites on the energy metabolism, migration, and proliferation of normal UCs, non-invasive RT4, and invasive T24 bladder cancer cells were established. The results demonstrated that chronic exposure to BPA-gluc or BPS-gluc decreased the metabolism and migration of normal UCs and increased their proliferation, resulting in a diminished wound-healing capacity. Exposure to BPA-gluc or BPS-gluc increased the bioenergetics and physiological activity of bladder cancer cells, particularly for non-invasive RT4 cells, which could promote the transition from a non-invasive to an invasive bladder cancer phenotype. This study demonstrates that BP metabolites are not as inactive as initially thought. Further studies on the impact of endocrine disruptors are required to understand better the role of BPs and their glucuronidated metabolites on bladder cancer development, since exposure to these compounds could have a significant clinical impact on bladder cancer progression and patients’ prognosis.

## Figures and Tables

**Figure 1 ijms-23-12859-f001:**
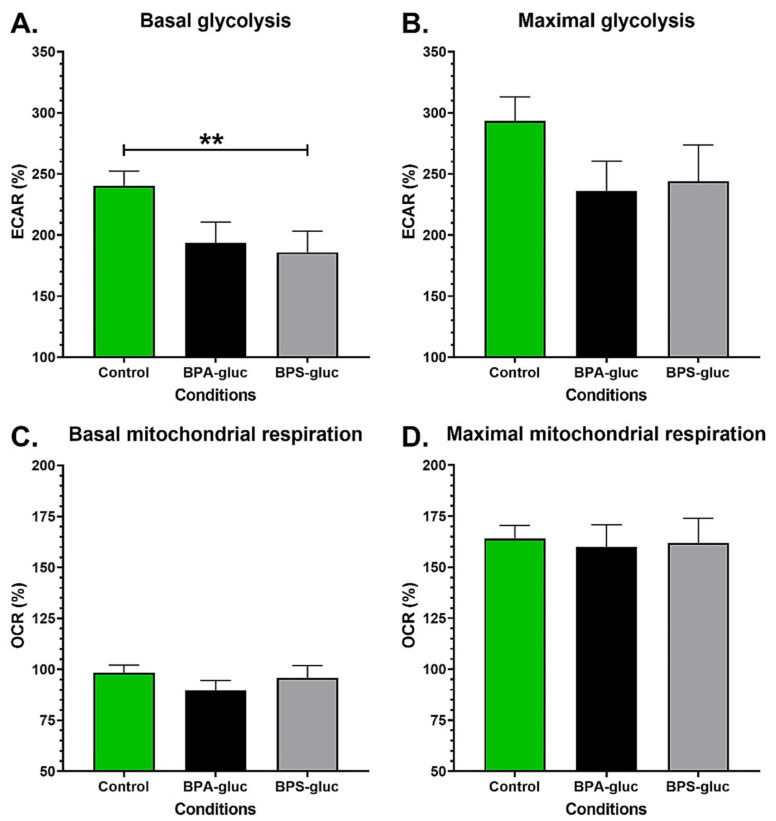
Chronic exposure to physiological concentrations of BPS glucuronidated metabolites decreases the basal glycolytic capacity of normal urothelial cells. (**A**,**B**) ECAR and (**C**,**D**) OCR were determined using the XFe96 Extracellular Flux Analyzer of normal urothelial cells (UCs) with or without chronic exposure to physiological concentrations of BPA-gluc or BPS-gluc to establish (**A**) basal glycolytic capacity, (**B**) maximal glycolytic capacity, (**C**) basal mitochondrial respiration and (**D**) maximal mitochondrial capacity. Analyses represent the combined results for two populations of normal urothelial cells (UC1 and UC2). Data are presented as the mean ± SEM and displayed as percentages of controls (i.e., untreated condition) (n ≥ 3, N = 4). The baseline (100%) was established before injections (see Appendix A). ** *p* < 0.01 by Mann–Whitney test.

**Figure 2 ijms-23-12859-f002:**
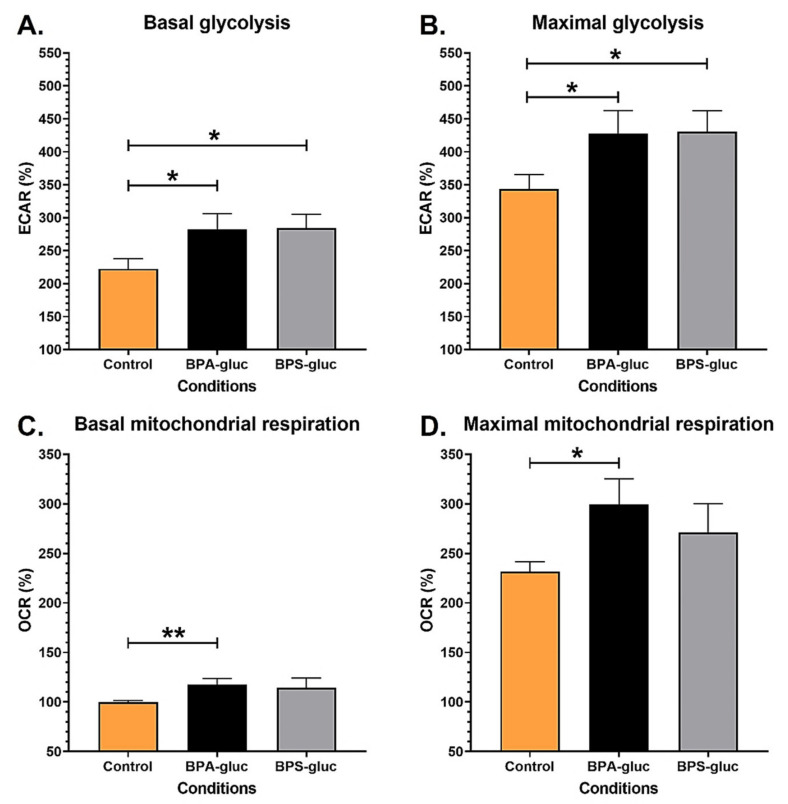
Chronic exposure to physiological concentrations of BPA and BPS glucuronidated metabolites increases the glycolytic and mitochondrial capacities of RT4 non-invasive bladder cancer cells. (**A**,**B**) ECAR and (**C**,**D**) OCR were determined using the XFe96 Extracellular Flux Analyzer of RT4 non-invasive bladder cancer cells with or without chronic exposure to physiological concentrations of BPA-gluc or BPS-gluc to establish (**A**) basal glycolytic capacity, (**B**) maximal glycolytic capacity, (**C**) basal mitochondrial respiration and (**D**) maximal mitochondrial capacity. Data are presented as the mean ± SEM and displayed as percentages of controls (i.e., untreated condition) (n ≥ 3, N = 3). The baseline (100%) was established before injections (see Appendix A). * *p* < 0.05, ** *p* < 0.01 by Mann–Whitney test.

**Figure 3 ijms-23-12859-f003:**
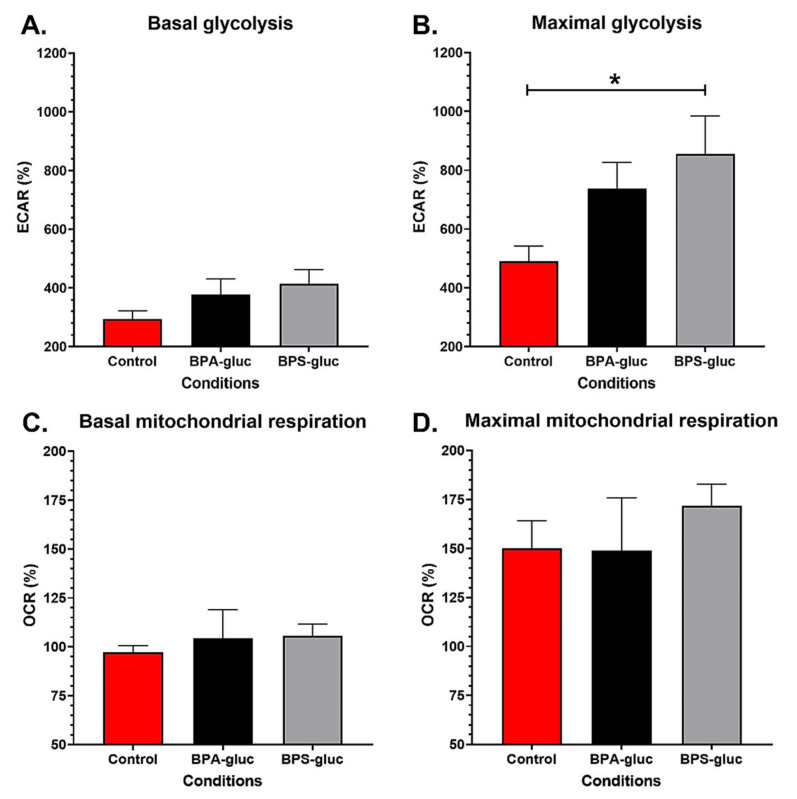
Chronic exposure to physiological concentrations of glucuronidated BPS increases the maximal glycolytic capacity of T24 invasive bladder cancer cells. (**A**,**B**) ECAR and (**C**,**D**) OCR were determined using the XFe96 Extracellular Flux Analyzer of T24 invasive bladder cancer cells with or without chronic exposure to physiological concentrations of BPA-gluc or BPS-gluc to establish (**A**) basal glycolytic capacity, (**B**) maximal glycolytic capacity, (**C**) basal mitochondrial respiration and (**D**) maximal mitochondrial capacity. Data are presented as the mean ± SEM and displayed as percentages of controls (i.e., untreated condition) (n ≥ 3, N = 3). The baseline (100%) was established before injections (see Appendix A). * *p* < 0.05 by Mann–Whitney test.

**Figure 4 ijms-23-12859-f004:**
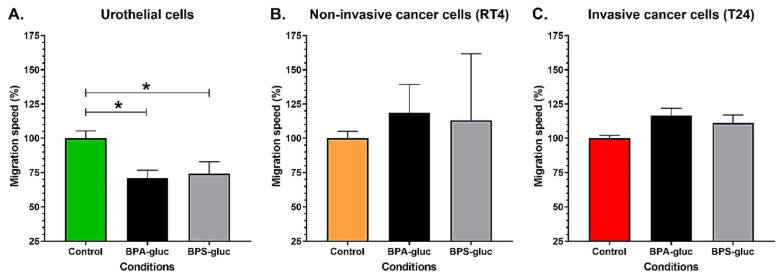
Chronic exposure to physiological concentrations of glucuronidated BPA or BPS decreases the migration speed of normal urothelial cells. The migration speed of (**A**) normal urothelial cells (UCs), (**B**) RT4 non-invasive bladder cancer cells, and (**C**) T24 invasive bladder cancer cells was evaluated by time-lapse microscopy with or without chronic exposure to physiological concentrations of BPA-gluc or BPS-gluc. Data are presented as the mean ± SEM and displayed as percentages of controls (i.e., untreated condition) (n = 2, N = 3). The 100% migration value of UCs’ control represents a mean of 12.1 cm^2^/h, for RT4 1.7 cm^2^/h, and for T24 4.8 cm^2^/h. * *p* < 0.05 by Mann–Whitney test.

**Figure 5 ijms-23-12859-f005:**
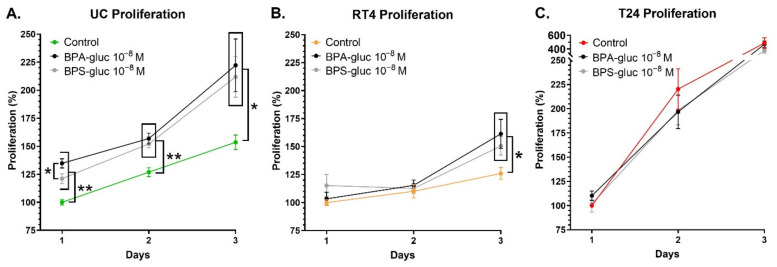
Chronic exposure to physiological concentrations of glucuronidated BPA and BPS increases the proliferation rate of UCs and RT4 non-invasive bladder cancer cells. The proliferation rate of (**A**) normal urothelial cells (UCs), (**B**) RT4 non-invasive bladder cancer cells, and (**C**) T24 invasive bladder cancer cells was established over three days, with or without chronic exposure to physiological concentrations of BPA-gluc or BPS-gluc. The proliferation rate is illustrated by the number of cells as a function of time. Data, presented as the mean ± SEM, are displayed as percentages of controls at day 1 (i.e., untreated condition) (n = 3, N = 3). * *p* < 0.05, ** *p* < 0.01 by Mann–Whitney test.

## Data Availability

The data presented in this study are available on request from the corresponding author.

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
