# Peer review of "Glucuronidated Metabolites of Bisphenols A and S Alter the Properties of Normal Urothelial and Bladder Cancer Cells"

_ijms, 2022, doi:10.3390/ijms232112859_

Round 1

Reviewer 1 Report

This paper reveals a public health issue that BPA and BPS materials are not so safe as people considered before. When BP metabolites are stored in bladder, they may promote bladder cancer progression. However, the work of this paper is too light, the results provided by authors cannot sufficiently support their conclusion. All the works authors did just detect superficial phenomena, there are not any detail mechanism studies in this paper.

1.     Figure 1, although author treated the cell with BPA-gluc or BPS-gluc in a physiological concentration, different concentrations treatment is also required, to make sure whether this phenotype changes have a dose-depend manner.

2.     Figure 5, I do not understand, as long-term treatment of BP metabolites decreases glycolysis in bladder normal urothelial cells, but why this treatment promotes normal urothelial cell proliferation. As we know glucose metabolism can provide materials for synthesis of lipid and DNA, which are essential for cell proliferation. Or whether curtain oncogenes are activated in this process.

3.     Figure 5, long-term treatment of BP metabolites increases glycolysis in T24, why this treatment failed to promote T24 proliferation? It is generally accepted that elevated glycolysis is a strategy cancer cell use to increase its proliferation.

4.     Figure 5, as T24 proliferation was not affected by BPA-gluc or BPS-gluc treatment, I am not sure whether BP metabolites can promote bladder cancer progression. Author should add at least one more bladder cancer cell line to repeat the experiment.

5.     Any cellular signaling pathways affected by BP metabolites.

6.     Metabolites profile and isotope tracing experiment by LC-MS must be done, to determine how BP metabolites affect metabolism in bladder normal cell and cancer cell.

Author Response

Grammar revision was performed by a native English-speaking colleague. Changes are not shown in the manuscript.

Reviewer 2 Report

Pellerin and co-authors explore the functional effects of glucuronidated metabolites of BPA and BPS on bladder normal and cancer cells. This topic is important for understanding the etiology of bladder cancer, especially because of the significant association of a genetic variation within the UGT region (encoding human glucuronidases) with bladder cancer risk (PMID 22228101). It is considered that glucuronides are stable but can dissociate at a lower pH of the urine which can be influenced by different conditions, thus releasing reactive metabolites. Could you comment on this in the discussion?

Minor: The Y-scale on the Figures represents % to controls (untreated conditions), but the controls bars are also shown – what are these?  Could you please clarify how these % should be interpreted? 

Reviewer 3 Report

In this article titled "Glucuronidated metabolites of bisphenols A and S alter the properties of normal urothelial and bladder cancer cells", the authors report the cellular effects of bisphenol metabolites on normal and cancerous urothelial/bladder cells. 

The manuscript is very precise. The introduction covers a good range of literature, justifying the study. Study is well designed and methodology is appropriate. Results are well written and fit with the purpose of the study. Detailed figure legends are included in the manuscript. Supplementary data is also provided.

Round 2

Reviewer 1 Report

As authors addressed my concerns and other reviewers have no further question, this paper could be accepted for publication.